# Risk Factors and the Impact of Multidrug-Resistant Bacteria on Community-Acquired Urinary Sepsis

**DOI:** 10.3390/microorganisms11051278

**Published:** 2023-05-13

**Authors:** Manuel Madrazo, Ian López-Cruz, Laura Piles, Sofía Viñola, Juan Alberola, José María Eiros, Arturo Artero

**Affiliations:** 1Hospital Universitario Doctor Peset, 46017 Valencia, Spain; manel.madrazo@gmail.com (M.M.); ilopezcruz5@gmail.com (I.L.-C.); laurapilesroger@gmail.com (L.P.); sofiavhernandez@gmail.com (S.V.); arturo.artero@uv.es (A.A.); 2Hospital Universitario Rio Hortega, 47012 Valladolid, Spain; eiros@med.uva.es

**Keywords:** risk factors, multidrug-resistant bacteria (MDRB), nosocomial urinary tract infection (UTI)

## Abstract

Risk factors for multidrug-resistant bacteria (MDRB) in nosocomial urinary tract infection (UTI) have been widely studied. However, these risk factors have not been analyzed in community-acquired urinary sepsis (US), nor have its outcomes been studied. The aim of our study is to determine risk factors for MDRB in community-acquired US and its influence on outcomes. Prospective observational study of patients with community-acquired US admitted to a university hospital. We compared epidemiological and clinical variables and outcomes of US due to MDRB and non-MDRB. Independent risk factors for MDRB were analyzed using logistic regression. A total of 193 patients were included, 33.7% of them with US due to MDRB. The median age of patients was 82 years. Hospital mortality was 17.6%, with no difference between the MDRB and non-MDRB groups. The length of hospital stay was 5 (4–8) days, with a non-significant tendency to longer hospital stays in the MDRB group (6 (4–10) vs. 5 (4–8) days, *p* = 0.051). Healthcare-associated US was found to be an independent risk factor for MDR bacteria by multivariate analysis. In conclusion, the impact of MDR bacteria on the outcomes of community-acquired urinary sepsis was mild. Healthcare-associated US was an independent risk factor for MDR bacteria.

## 1. Introduction

Multidrug-resistant bacteria (MDRB) have a significant clinical and economic burden and has become a major healthcare issue worldwide [1,2]. MDRB commonly associated with nosocomial infections and their prevalence has increased in recent years [3]. Moreover, they are becoming a significant prevalent cause of community-acquired infections too [4]. International bodies have established as a priority objective to advance in the understanding and dissemination of new knowledge on antibiotic resistance, as well as decrease both healthcare-associated and community-acquired antibiotic-resistant infections [5].

Urinary tract infection (UTI) is one of the most frequent causes of bacteremia and sepsis [6,7], and *Enterobacteriaceae* are the main causes of UTI. Extended-spectrum beta-lactamases (ESBL) are a resistance mechanism that are increasingly common among these bacteria. ESBL are a version of beta-lactamase enzymes that hydrolyze additional beta-lactam antibiotics, including penicillins, many cephalosporins and aztreonam, affecting the effectiveness of the majority of beta-lactams but not carbapenems [8]. Development of other resistance mechanisms is frequent since they are often encoded in the same plasmids that contain the ESBL genes [9,10]. Resistance to fluoroquinolones is multifactorial and can be via one or a combination of target-site gene mutations, increased production of efflux pumps, modifying enzymes, and/or target-protection proteins [11]. Resistance to fluoroquinolones that have the same risk factors as mechanisms of resistance to other antibiotics [12] and it has been found that an association between resistance to fluoroquinolones and both MDRB [13] and ESBL-producing *Enterobacteriaceae* in UTIs has been found [14]. That is why MDRB are an increasing cause of UTI both in community-acquired and healthcare-associated infections [15,16,17], which might increase the rate of treatment failure [17]. *Pseudomonas aeruginosa* and *Enterococcus* spp. are also frequent causes of UTI, especially in healthcare-associated infections [18]. Antimicrobial resistance among these microorganisms is also increasing and making them become a difficult-to-treat cause of infection. Several mechanisms of antibiotic resistance are described in *P. aeruginosa*. The most important is reduced permeability, active efflux and degrading enzymes (including ESBLs), which frequently coexist in MDR isolates [19]. Enterococci have a variety of intrinsic mechanisms of antibiotic resistance, and *E. faecium* can present with high-level resistance to ampicillin and vancomycin with a major impact on therapeutic options [20].

Risk factors for MDRB in nosocomial UTI and community-acquired sepsis from different sources have been widely studied. Between the most important risk factors, we can find the presence of an indwelling catheter, antibiotic treatment within the previous 30 days, a prior episode of UTI within the previous year [21], hospitalization in the previous 90 days, septic shock, previous stroke [22], and long-term care facility residency [21,23]. However, these risk factors have not been specifically analyzed in community-acquired urinary sepsis, nor have their outcomes been studied.

The aim of our study is to determine the risk factors for multidrug-resistant bacteria in community-acquired urinary sepsis and its influence on outcomes to improve the treatment of our patients.

## 2. Material and Methods

### 2.1. Study Design and Patients

Prospective observational study of patients admitted to a university hospital diagnosed with community-acquired urinary sepsis from January 2017 to June 2022. Cases with a non-positive urine culture, or with a discharge diagnosis of asymptomatic bacteriuria with a clinical syndrome compatible with any other condition after reviewing the case were excluded, as well as nosocomial cases of UTI and cases of UTI transferred from the Intensive Care Unit (Figure 1). Epidemiological, clinical, and microbiological variables were collected by the authors in an ad hoc form. All the cases were reviewed by two independent researchers on ward in order to rule out any other sources of infection before being included in the study. In the event of any disagreement, the case was assessed by a third investigator.

### 2.2. Data Collection and Definitions

All patients included in this study met the diagnostic features of complicated urinary tract infection, namely: pyuria and bacteriuria on urinalysis by microscopy and at least one of the following: (i) symptoms of cystitis (dysuria, urinary urgency, and/or urinary frequency) along with fever or other symptoms of systemic illness such as chills, rigors, or acute mental status changes; (ii) flank pain and/or costovertebral tenderness; or (iii) fever or sepsis without localizing symptoms if other causes have been ruled out. Some cases were excluded when, despite meeting these criteria, the urine culture was negative in the following days.

Multidrug resistance (MDR) was defined according to the international expert proposal by Magiorakos et al. [24], as non-susceptibility to at least one agent in three or more antimicrobial categories (extended-spectrum penicillins, carbapenems, cephalosporins, aminoglycosides, and fluoroquinolones for Gram-negative bacteria; and ampicillin, vancomycin, fluoroquinolones, fosfomycin, and linezolid for Gram-positive bacteria). Extensively drug-resistance (XDR) was defined as non-susceptibility to at least one agent in all but two or fewer antimicrobial categories tested for a particular microorganism. The antibiotics family considered for resistance were according to the recommendations of the Clinical and Laboratory Standards Institute [25]. In the case of polymicrobial UTI, we considered the microorganism with the most resistances for the purposes of classification in the group of MDRB and inadequate empiric antimicrobial therapy (IEAT), but the rate of resistance was considered by microorganisms.

Sepsis was defined according to sepsis-3 criteria [26] and identified as an acute change in total SOFA score ≥ 2 points consequent to the infection. SOFA and quick SOFA (qSOFA) scales were used according to their original definitions [27,28]. Blood pressure, respiratory rate and peripheral oxygen saturation, Glasgow Coma Scale, diuresis, use of vasopressors drugs and laboratory analysis with renal function, bilirubin, and hemogram, among others, were measured within the first 24 h of admission at the Emergency Department. Charlson comorbidity index and Barthel index for Activities of Daily Living were calculated to identify high comorbidity and severe dependency, respectively.

Community onset healthcare-associated (HCA) urinary sepsis was defined as a community-onset urinary sepsis with any of the following criteria: (i) to have been admitted to an acute care hospital ≥ 48 h within 90 days prior to current hospital admission; (ii) to have received antimicrobial therapy within 90 days prior to admission; and (iii) residing in a nursing home [29]. Community-acquired infection was defined when the symptoms of the urinary infection were initiated in the community and none of the previous criteria were met [29].

IEAT was considered as the occurrence of infection that was not effectively treated at the time when the causative microorganism and its antimicrobial susceptibility were known. This included the absence of antimicrobial agents directed at a specific class of microorganisms and the administration of an antimicrobial agent to which the microorganism responsible for the infection was resistant [30].

### 2.3. Statistical Analysis

Quantitative variables were summarized with means and standard deviation or medians and interquartile ranges (IQR) according to their distribution and compared by using Student’s t-test or analysis of variance (ANOVA) when the distribution was normal, or Mann–Whitney U-test when it was not normal. Qualitative variables were expressed as absolute and relative (percentage) frequencies and compared with chi square test or Fisher’s exact test. Risk factors for MDRB (Table 1) and 30-day mortality were identified in the univariate analysis and were included in the multivariate analysis, which was performed using logistic regression (Table 2). In the table of etiology (Table 3), chi square test was used in groups with n ≥ 30, while Fisher’s exact test was used in groups with n < 30; groups of less than 5 bacteria were not analyzed. Outcomes (Table 4) were analysed using chi square test or Mann–Whitney U-test, as appropriate. In Table 5, the univariate analysis was conducted with the chi square test, while the multivariate analysis was conducted by logistic regression. All tests were two-tailed and an α significance level of 0.05 was considered to show statistical significance. The statistical package SPSS version 22 from IBM for Windows [International Business Machines, Armonk, New York, NY, USA] was used for the statistical analysis.

## 3. Results

Out of the 1228 cases of UTI diagnosed on admission, 193 patients with urinary sepsis were included, 33.7% of them with urinary sepsis due to MDRB (Figure 1). Genders were almost equally distributed (50.8% females) and median age was 82 years, with 95.3% of the patients with high comorbidity (Charlson ≥ 3). Diabetes mellitus (37.8%), chronic kidney disease (35.8%), and dementia (33.7%) were the three more frequent comorbidities, with no significant differences between groups (Table 1).

*Escherichia coli* (52.6%), *Klebsiella pneumoniae* (13.9%), *Pseudomonas aeruginosa* (7.9%), and *Enterococcus faecalis* (6.5%) were the most frequent microorganisms isolated in the urine cultures, with no differences between the MDRB group and non-MDRB groups (Table 2), although ESBL-producing *Enterobacteriaceae* were related to the MDRB group. Other *Enterobacteriaceae* than *E. coli* and *K. pneumoniae* showed a tendency to be related to the MDRB-urinary sepsis group, as well as *P. aeruginosa* and *E. faecalis* tended to be related to the non-MDRB-urinary sepsis group, but neither of them were statistically significant. The three cases of candiduria were in the context of polymicrobial UTI. In the multidrug-resistant group, 60 cases were MDR and 5 cases were XDR, corresponding to 71 MDR microorganisms and 5 cases of XDR microorganisms. There were no cases of pandrug-resistant bacteria.

Resistance to ciprofloxacin (total 36.8%, 66.2% vs. 21.6% in the MDRB and non-MDRB groups, respectively, *p* < 0.001) and resistance to third-generation cephalosporins (total 20.1%, 49.2% vs. 2.9% in the MDRB and non-MDRB groups, respectively, *p* < 0.001) were associated with the MDRB group. There were only 3 cases of resistance to carbapenems, two of them in the MDRB group, and the other in the non-MDRB group. *Enterococcus faecalis* was the most frequent Gram-positive bacteria (6.5% of the cases), with no cases of resistance to vancomycin, although there were two cases of MDR *E. faecalis*. Both cases were older patients with HCA-urinary sepsis and polymicrobial UTI, and they were resistant to quinolones, fosfomycin, and ampicillin, but not to vancomycin.

Ceftriaxone (43.5%), meropenem (23.8%), and amoxicillin-clavulanate (7.8%) were the most frequent empirical treatment prescribed on admission. Analyzing the different groups of empiric antimicrobial treatment, non-carbapenems beta-lactam antibiotics in monotherapy were the most frequent group with 52.8% of the cases, followed by carbapenems with 30.1% of the cases. The most frequent combined therapy was a beta-lactam plus gentamicin (5.7% of the cases), there were no differences between the MDRB and the non-MDRB groups, except for the anti-pseudomonal beta-lactams subgroup (9.2% vs. 1.6% in the MDRB and non-MDRB groups, respectively, *p* = 0.012), but it only accounted for 4.1% of all the antimicrobials empirically prescribed. Empirical antimicrobial treatment was changed because of the clinical evolution or adjusted according to antibiogram in the first 24 h after admission in 43.5% of the cases (50.8% vs. 40.8% MDRB and non-MDRB groups, respectively, *p* = 0.148). The IEAT was 23.8%, also without differences between both groups (27.7% vs. 21.9% in the MDRB and non-MDRB groups, respectively, *p* = 0.370).

In-hospital mortality was 17.6% and 30-day mortality was 23.3% with no difference between the MDRB and non-MDRB groups, while length of hospital stay was 5 (4–8) days, with a non-significant tendency to longer hospital stay in the MDRB group (6 (4–10) vs. 5 (4–8) days, *p* 0.051) (Table 4).

Severe dependency, altered mental status, and healthcare-associated urinary sepsis were associated with urinary sepsis due to MDRB in the univariate analysis (Table 1). In the multivariate analysis, healthcare-associated urinary sepsis was found to be an independent risk factor for urinary sepsis due to MDRB (Table 2).

Because of these findings, we decided to analyze the influence of healthcare-associated urinary sepsis. HCA-urinary sepsis was associated with MDRB (45.5% vs. 18.1% in the HCA-US and non-HCA-US groups, respectively, *p* < 0.001), but also with IEAT (30% vs. 15.7%, respectively, *p* = 0.021), ESBL-EB (15.5% vs. 2.4%, respectively, *p* = 0.003), urinary sepsis due to *Pseudomonas aeruginosa* (12.7% vs. 3.6%, respectively, *p* = 0.027), and resistance to fluoroquinolones (49.5% vs. 19.8%, respectively, *p* < 0.001) and third-generation cephalosporins (29.3% vs. 8.3%, respectively, *p* = 0.001) in the univariate analysis. However, only MDRB (*p* = 0.025, OR 2.9 (95% CI 1.1–7.4) and resistance to fluoroquinolones (*p* = 0.025, OR 2.6 (95% CI 1.2–6.2) were associated to HCA-urinary sepsis in the multivariate analysis by logistic regression.

Known risk factors for 30-day mortality, such as age, comorbidity, dependency, HCA-urinary sepsis, septic shock and IEAT [31,32,33,34,35], were analyzed by univariate analysis (Table 5). Barthel ≤ 40, HCA-urinary sepsis, and septic shock had statistically significant differences between the 30-day mortality group and the survivors in the univariate analysis. A multivariate analysis by logistic regression of these statistically significant variables were analyzed, along with urinary sepsis due to MDRB, although there was no statistically significant relationship in the univariate analysis. Severe dependency (Barthel ≤ 40, *p* < 0.001, OR 4.1 (95% IC 1.8–8.9) and septic shock (*p* = 0.006, OR 3.2 (95% IC 1.4–7.1)) were related to 30-day mortality in the multivariate analysis.

## 4. Discussion

This study shows that the impact of MDRB in patients with community-acquired urinary sepsis was moderate. There were no differences in in-hospital mortality or 30-day mortality between the MDRB group and non-MDRB group. Although there was a tendency for a longer hospital stay in the MDRB group, this difference was not statistically significant.

Community-acquired urinary sepsis was caused by MDRB in 33.7% of the cases, comparable to other studies in similar settings (34.1% to 36.5%) [12,36,37]. It is higher than in other studies on UTI outpatients (1.6% to 1.9%) [16,38] or in the Emergency Department (6.7%) [39], but lower than that found in another multicentric study conducted in nursing homes (46.1%) [40], which is consistent with our findings that correlate MDRB with HCA-urinary sepsis, one of whose variables is residing in a nursing home.

HCA-urinary sepsis was related to MDRB as etiology of community-acquired urinary sepsis in the multivariable analysis, highlighting the importance of a thorough anamnesis. Prior antimicrobial therapy is one of the most important variables in relation to the development of resistance mechanisms [12,41,42], as well as institutionalization in a nursing home [12,39,41,42,43,44]. Previous hospitalization has been related to resistance to some antimicrobials in some studies [12,41,44], but not in others [39,42,43]. Use of indwelling catheter was not related to MDRB in this study, as well as in others [42,43], although it was related to resistance in some other studies [12,45].

Severe dependency (Barthel < 40) and altered mental status on admission were not associated with MDRB, as in other studies with younger patients [42,45]. In this study older age, male sex or comorbidities were not associated with MDRB, in consonance with other studies [39,45] and contrary to other studies [41,46].

As expected in community-acquired UTI, the main microorganisms found in the urine culture were *E. coli* (52.6%), *K. pneumoniae* (13.9%), *P. aeruginosa* (7.9%), *E. faecalis* (6.5%), and *P. mirabilis* (5.6%) [47,48]. In the MDRB-urinary sepsis group, the main pathogens were also *E. coli* and *K. pneumoniae*, followed by *P. mirabilis*, similar to other studies on bacteraemic UTI due to *Enterobacteriaceae* [46] and complicated UTI caused by Gram-negative bacteria [21]. However, in this last study [21], which included community-acquired, nosocomial, and nursing home-acquired UTI, *P. aeruginosa* was also one of the main microorganisms (9% of the cases, 38.5% of them being MDRB), while in this study a similar percentage of cases (7.9%) were due to *P. aeruginosa*, but only 17.6% of them were MDRB. The different origin of the patients and the known relationship between nosocomial UTI and *P. aeruginosa* [49], as well as nursing home residence [50,51], might explain this difference.

A 13.4% of *Enterobacteriaceae* producing ESBL was described by Smithson et al. [42] in UTI in older patients in the Emergency department, similar to the 11% described in this study, being both lower than the 16.9% to 20% described in studies in North America and Europe [52,53], which included patients with both community-acquired and nosocomial UTI. Our results were higher than a nationwide cross-sectional French study of community-acquired *E. coli* UTI [54], in which the *E. coli* producing ESBL were 3% of the cases, although this study included children and outpatients, and the 5% ESBL-EB observed in community-acquired UTI by Horcajada et al. [47].

Resistance to ciprofloxacin was 36.8%, higher than the rate of quinolone resistance ranged between 23 and 31% shown in other studies conducted in Europe and North America [12,55,56,57], which included community-acquired uncomplicated UTI, complicated UTI, and pyelonephritis; however, it is lower than other studies conducted in Europe, with a 47.2% rate of resistance to ciprofloxacin in community-acquired complicated UTI [48]. In any case, the high rates of quinolone resistance warrant not using fluoroquinolones as empirical antimicrobial therapy [58,59], that supports the low use of fluoroquinolones as empirical treatment in our study (4.1%). In this study, resistance to third-generation cephalosporins was 20.1%, similar to other European studies with community-acquired complicated UTI (19.6%) [48]. Both ciprofloxacin and third-generation cephalosporines are associated with MDRB urinary sepsis. Interestingly, when we analyzed the association with the HCA-urinary sepsis, both resistance patterns are related to HCA-urinary sepsis; however, while the resistance to ciprofloxacin is close to 20% in the non-HCA urinary sepsis and, therefore, not advisable as empirical treatment [58,59], the resistance to third-generation cephalosporines is 8.3% in the non-HCA urinary sepsis group, allowing us to assess its use as empirical treatment in these cases in which the conditions of HCA-urinary sepsis are not present.

Vancomycin-resistant enterococci (VRE) have been reported to be a significant nosocomial microorganism all over the world [60], with vancomycin-resistant *Enterococcus faecium* being the leading MDR enterococci in healthcare environments [61], with a rising VRE rate in Europe from 8.1% in 2012 to 19% in 2018 [62]. It was not the case in this study, in which the two isolates of *E. faecium* were non-MDRB, sensitive to ampicillin and vancomycin. While vancomycin-resistant *E. faecalis* has been reported to be of increasing its prevalence worldwide, in Europe, it has remained generally susceptible to vancomycin [63], with a stable rate of vancomycin-resistant *E. faecalis* of 1.1% during the period 2012–2018 [62]. This is consistent with the results of this study, in which none of the *E. faecalis* were resistant to vancomycin, although there were two cases of MDR *E. faecalis.* While MDR *E. faecalis* is normally associated with high resistance to vancomycin [64,65], it was not the case in our study.

In-hospital mortality for sepsis from urinary focus has been described as 40% in a large multicentric retrospective study carried out from 2004 to 2018 in the USA [66], which included patients from the ICU, medical wards, nursing homes, and outpatients, which could explain why it is higher than the results in this study, 17.6% and 23.3% for in-hospital and 30-day mortality, respectively. Tocut et al. [67] described a 30-day mortality of 8.6% in their retrospective cohort; however, their diagnosis of urinary sepsis was only based on clinical criteria and their patients were younger. Holmbom et al. [31] described a 30-day mortality of 14% in a retrospective analysis of an open urologic cohort in Sweden; however, diagnosis of sepsis was clinical (ICD-10 codes), the mean age of the cohort was a decade younger than the mean age in this study, and the mean Charlson Comorbidity Index was 2.2, while in this study 95.3% of the patients had a Charlson Comorbidity index of ≥3.

Our results suggest that MDRB does not necessarily have a deleterious effect, as shown in other works [68,69]. IEAT was not related to mortality, as shown in another study with bloodstream infection [33] with 31.4% of patients with a UTI. However, it has been related to mortality in patients with UTI [31,34]. According to the results of this study, with patients admitted to a hospital ward with urinary sepsis, the clinical severity at admission (septic shock) is what determines the patient’s prognosis in the urinary sepsis [31], not the etiology due to MDRB or non-MDRB.

It should be noted that in this study we found no difference in the IEAT between the groups of MDRB and non-MDRB, contrary to another previous study of our group [14] with community-acquired UTI in older patients. This may be because in the present study, due to more severe clinical, empirical treatment with carbapenems was started in the Emergency Department in a higher percentage (25.9% in this study, 21.4% in our previous study). It may also be due to the faster and higher percentage of determination of the etiology and antibiogram in UTI, compared to other frequent infections, such as respiratory infections or skin and soft tissue infections [70,71]. This allows an early adjustment of antibiotic therapy, making it possible to reduce the effect of IEAT on mortality [33,40].

For this reason, we consider that, although in this study the etiology of MDRB does not seem to determine the prognosis, in UTI in other circumstances or in other infections in which the isolation of the microorganisms is not so frequent, it may be very important to know the risk factors for MDRB. This would allow us to carry out an early adjustment of antibiotic therapy, making it possible to reduce the effect of IEAT on mortality.

The main strength of our study is its clinical conception, based on the information that we can obtain from the patient through the anamnesis and physical examination, not focusing on a specific group of MDRB such as *Enterobacteriaceae* [72] or *Pseudomonas aeruginosa* [45]. We believe this approach may help the clinicians to better treat the patients they encounter in the Emergency Department or in the hospital ward. Another of the strengths of our study is its prospective design and the rigorous selection of cases, as can be seen in the number of excluded cases (Figure 1).

The primary limitation of this work is that it was carried out in a single center, with the bias that this may mean for the rate of resistance and the empirical choice of antibiotics. This study was carried out with community-acquired urinary sepsis; therefore, the results may not be extrapolated to other populations, such as nosocomial infections or patients residing in a nursing home. Another limitation is that we did not collect the history of previous UTI or MDRB infection, which has been considered a risk factor for antibiotic resistance [53,73]. Finally, the older age of the study participants may be a limiting factor for direct comparison to other studies carried out on younger patients.

## 5. Conclusions

In conclusion, patients with MDRB in our study had similar rates of mortality and hospitalization compared to non-MDRB patients, with healthcare-associated US being noted to be an independent risk factor for MDR bacterial infection.

## Figures and Tables

**Figure 1 microorganisms-11-01278-f001:**
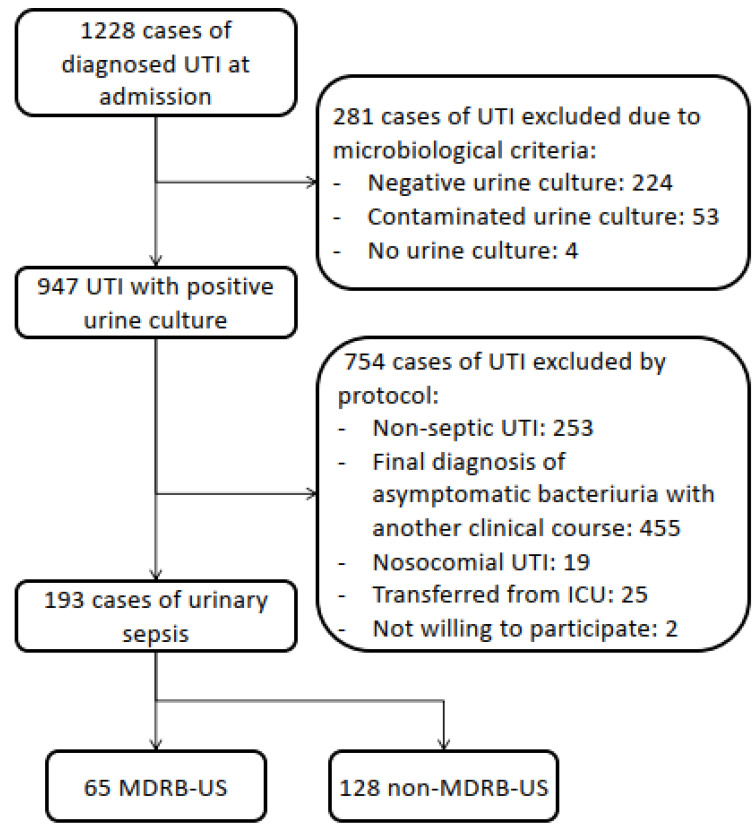
Flowchart of inclusion of 193 cases of community-acquired urinary sepsis.

**Table 1 microorganisms-11-01278-t001:** Epidemiological and clinical characteristics and outcomes of community-acquired urinary sepsis due to multidrug-resistant bacteria and non-multidrug-resistant bacteria.

	Total N 193	MDRB-US N 65 (33.7%)	Non-MDRB US N 128 (66.3%)	*p*
Female sex, n (%)	98 (50.8)	31 (47.7)	67 (52.3)	0.541
Age (years), median (IQR)	82 (76–88)	84 (77–89)	82 (76–88)	0.305
Charlson ≥ 3, n (%)	184 (95.3)	62 (95.4)	122 (95.3)	0.982
Barthel < 40, n (%)	85 (44)	38 (58.5)	47 (36.7)	**0.004**
**Comorbidities**				
Dementia, n (%)	65 (33.7)	25 (38.5)	40 (31.3)	0.208
Diabetes mellitus, n (%)	73 (37.8)	29 (44.6)	44 (34.4)	0.166
COPD, n (%)	24 (12.4)	7 (10.8)	17 (13.4)	0.604
CKD, n (%)	69 (35.8)	25 (38.5)	44 (34.6)	0.602
Cancer, n (%)	40 (20.7)	13 (20)	27 (21.1)	0.859
Indwelling urinary catheter, n (%)	36 (18.7)	16 (24.6)	20 (15.6)	0.130
HCA-UTI, n (%)	110 (57)	50 (76.9)	60 (46.9)	**<0.001**
Previous hospitalization, n (%)	66 (34.2)	31 (47.7)	35 (27.3)	**0.005**
Previous antimicrobial therapy, n (%)	89 (46.1)	37 (56.9)	52 (40.6)	**0.032**
Nursing home residence, n (%)	17 (8.8)	13 (20)	4 (3.1)	**<0.001**
**Clinical characteristics**				
APACHE II, median (IQR)	15 (11–20)	16 (13–21)	15 (11–19)	0.117
APN, n (%)	109 (56.5)	38 (58.5)	71 (55.5)	0.692
Altered mental status, n (%)	120 (62.2)	49 (75.4)	71 (55.9)	**0.008**
RR ≥ 22 bpm, n (%)	76 (39.4)	23 (35.4)	53 (41.7)	0.395
SBP < 100 mmHg, n (%)	66 (34.2)	25 (38.5)	41 (32.3)	0.394
Fever, n (%)	142 (73.6)	52 (80)	90 (70.3)	0.149
qSOFA ≥ 2, n (%)	104 (53.9)	40 (61.5)	64 (50)	0.129
Septic shock-3, n (%)	39 (20.2)	15 (23.1)	24 (18.8)	0.479
Lactate ≥ 2 mg/dl, n (%)	106 (54.9)	34 (52.3)	72 (56.3)	0.603
Leukocytosis, median (IQR)	13,900 (10,050–19,050)	13,900 (10,700–20,500)	13,700 (9650–18,750)	0.281
Blood cultures positive/BC taken (%)	55/125 (44)	17/40 (42.5)	38/85 (44.7)	0.632

Urinary sepsis due to multirresistant-bacteria, MDRB-US; COPD, chronic obstructive pulmonary disease; CKD, chronic kidney disease; HCA-UTI, healthcare associated-UTI; APN, acute pyelonephritis; RR, respiratory rate; SBP, systolic blood pressure; BC, blood cultures.

**Table 2 microorganisms-11-01278-t002:** Multivariate analysis of risk factors for community-acquired urinary sepsis due to multidrug-resistant bacteria.

	Univariate Analysis *p*	Multivariate Analysis *p*	OR (95% CI)
Barthel < 40	0.004	0.180	1.6 (0.8–3.1)
Healthcare-associated urinary sepsis	<0.001	0.001	3.1 (1.6–6.2)
Altered mental status	0.008	0.121	1.7 (0.9–3.7)

**Table 3 microorganisms-11-01278-t003:** Etiology of 193 cases of community-acquired urinary sepsis in hospitalized patients according to multidrug-resistant bacteria and non-multidrug-resistant bacteria.

	Total N 215	MDRB N 76 (35.3)	Non-MDRB N 139 (64.7)	*p*
**Gram-negative bacteria**, n (%)				
*Escherichia coli*	113 (52.6)	36 (47.4)	77 (55.4)	0.525
*Klebsiella pneumoniae*	30 (13.9)	12 (15.8)	18 (12.9)	0.425
*Klebsiella oxytoca*	6 (2.7)	4 (5.3)	2 (1.4)	0.082
*Proteus mirabilis*	12 (5.6)	7 (9.2)	5 (3.6)	0.062
Other *Enterobacteriaceae*	12 (5.6)	9 (11.8)	3 (2.1)	0.059
ESBL-EB *	19 (11)	17 (25)	2 (1.9)	**<0.001**
*Pseudomonas aeruginosa*	17 (7.9)	3 (3.9)	14 (10.1)	0.143
*Acinetobacter baumanii*	1 (0.5)	1 (1.3)	0	-
**Gram-positive bacteria**, n (%)				
*Enterococcus faecalis*	14 (6.5)	2 (2.6)	12 (8.6)	0.111
*Enterococcus faecium*	2 (0.9)	0	2 (1.4)	-
*Enterococcus gallinarum*	2 (0.9)	1 (1.3)	1 (0.7)	-
*Streptococcus agalactiae*	2 (0.9)	0	2 (1.4)	-
*Staphylococcus aureus*	1 (0.5)	0	1 (0.7)	-
**Fungi**, n (%)				
*Candida* spp.	3 (1.4)	1 (1.3)	2 (1.4)	-
**Polymicrobial US**, n (%)	20 (10.4)	9 (13.8)	11 (8.6)	0.258

MDRB, multidrug-resistant bacteria; US, urinary sepsis; ESBL-EB, *Enterobacteriaceae* producing extended-spectrum beta-lactamases. * Percentage of ESLB-EB was calculated including only the cases caused by *Enterobacteriaceae* (n = 173).

**Table 4 microorganisms-11-01278-t004:** Outcomes of community-acquired urinary sepsis due to multidrug-resistant bacteria and non-multidrug-resistant bacteria.

	Total N 193	MDRB-US N 65 (33.7%)	Non-MDRB US N 128 (66.3%)	*p*
In-hospital mortality, n (%)	34 (17.6)	13 (20)	21 (16.4)	0.536
30-day mortality, n (%)	45 (23.3)	19 (29.2)	26 (20.3)	0.166
Length of hospital stay (days), median (IQR)	5 (4–8)	6 (4–10)	5 (4–8)	0.051

Urinary sepsis due to multidrug-resistant bacteria, MDRB-US.

**Table 5 microorganisms-11-01278-t005:** Univariate and multivariate analysis by logistic regression of risk factors for 30-day mortality of 193 cases of community-acquired urinary sepsis HCA-US, healthcare associated-urinary sepsis; IEAT, inadequate empiric antimicrobial therapy; MDR-US, urinary sepsis due to multidrug-resistant-microorganisms.

	Univariate Analysis *p*	Multivariate Analysis *p*	OR (95% CI)
Age ≥ 75 years	0.482	-	
Charlson ≥ 3	0.090	-	
Barthel ≤ 40	<0.001	<0.001	4.1 (1.8–8.9)
HCA-US	0.004	0.074	2.1 (0.9–4.9)
Septic shock	0.001	0.006	3.2 (1.4–7.1)
IEAT	0.611	-	
MDR-US	0.166	0.917	0.9 (0.4–2.1)

## Data Availability

The data presented in this study are available on request from the corresponding author.

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
