# Peer review of "Risk Factors and the Impact of Multidrug-Resistant Bacteria on Community-Acquired Urinary Sepsis"

_microorganisms, 2023, doi:10.3390/microorganisms11051278_

Round 1

Reviewer 1 Report

Dear Authors, 

The article raises a serious problem of sepsis-causing infections. Detection of factors that increase the risk of such a situation is very important from the point of view of clinicians and patients themselves. However, the risk factors detected in the work are not very surprising, although they should be kept in mind. Regardless, the work requires significant corrections.

Below are the comments:

1. Abstract: there should be no abbreviations, some sentences are ungrammatical (e.g. "Prospective observational study of patients with community-acquired US admitted to a university hospital.")

2. The aim of the study is unclear, please explain, did  you look for the risk factors for the development of multidrug resistance in bacteria that cause sepsis, or risk factors for infection with MDR strains?

3. Line 78-79:  is unclear, please clarify what parameters were decisive for the inclusion of a given case in the study and why a three reviewers were  required, were there no clearly defined guidelines? Please explain or change.

4. please complete the information what statistical tests were used in tables.

5. Line 165: change to: Beta-lactam antibiotics.

6. There are a lot of brackets in the results, which make it difficult for the reader to understand the described results, please change it somehow, not always all values from the tables should be placed in the text.

7. line 216: change to: statistically significant

8. The discussion gives the impression of re-reading the results, please edit this part to avoid repetition of results, especially numbers, and focus on their interpretation and conclusions.

9. Line 298: how do authors know that MDRB strains are not more virulent than others? They did not test virulence, please remove this statement or explain on what basis they assessed virulence?

10.  Line 315 - 320:: A sentence with 6 lines and a whole paragraph is way too long and hard to understand. Please rephrase it.

Author Response

Reviewer 1

Thank you to the reviewers for their comments and suggestions, which have been most helpful and useful for improving the quality of the text and the readers’ comprehension. I will answer the reviewers’ questions point by point.

  1. Abstract: there should be no abbreviations, some sentences are ungrammatical (e.g. "Prospective observational study of patients with community-acquired US admitted to a university hospital.")

1 Answer. Abbreviations have been removed from the abstract. Thank you for your suggestion. This sentence has been corrected and the reviewed sentence is: “Prospective observational study on patients with community-acquired US admitted to a university hospital." (lines 12-13)

  1. The aim of the study is unclear, please explain, did you look for the risk factors for the development of multidrug resistance in bacteria that cause sepsis, or risk factors for infection with MDR strains?

2 Answer. The aim of the study is to determine the risk factors for infection caused by MDR strains. Therefore, we have changed a sentence in the abstract which read “The aim of our study is to determine risk factors for MDRB in community acquired US and its influence on outcomes (pag. 1, lines 11-12). The new sentence reads - “The aim of our study is to determine risk factors for community acquired US caused by MDRB and its influence on outcomes”.

  1. Line 78-79:  is unclear, please clarify what parameters were decisive for the inclusion of a given case in the study and why a three reviewers were required, were there no clearly defined guidelines? Please explain or change.

3 Answer. In some cases, patients were admitted to hospital where urinary infection was initially diagnosed at the emergency department because they had symptoms of infection and the presence of bacteria in urine. Before these cases were included in the study, they were reviewed by two researchers on ward in order to rule out any other sources of infection. In the event of any disagreement the case was assessed by a third investigator. You can see in Figure 1 that 455 cases were not included in the study because they have a final diagnosis of asymptomatic bacteriuria with another clinical course.

The following sentence has been amended in line 76-79: “All the cases were reviewed by two independent researchers on ward in order to rule out any other sources of infection before being included in the study. In the event of any disagreement the case was assessed by a third investigator.”

  1. please complete the information what statistical tests were used in tables.

Answer 4. The information has been added to the statistical analysis in material and methods:

Quantitative variables were summarized with means and standard deviation or medians and interquartile ranges (IQR) according to their distribution and compared by using Student's t-test or analysis of variance (ANOVA) when the distribution was normal, or Mann-Whitney U-test when it was not normal. Qualitative variables were expressed as absolute and relative (percentage) frequencies and compared with chi square test or Fisher’s exact test. Risk factors for MDRB (Table 1) and 30-day mortality were identified in the univariate analysis and were included in the multivariate analysis, which was performed using logistic regression (Table 4). In the table of etiology (Table 2), chi square test was used in groups with n≥30, while Fisher’s exact test was used in groups with n <30; groups of less than 5 bacteria were not analysed. Outcomes (Table 3) were analysed using chi square test or Mann-Whitney U-test, as appropriate.

A new table has been included in the manuscript (Table 5), following the another reviewer's  suggestion. In Table 5, the univariate analysis was done with the chi square test, while the multivariate analysis was done by logistic regression. All tests were two-tailed and an α significance level of 0.05 was considered to show statistical significance. The statistical package SPSS version 22 from IBM for Windows [International Business Machines, Armonk, New York, USA] was used for the statistical analysis.

  1. Line 165: change to: Beta-lactam antibiotics.

Answer 5. It has been done.

  1. There are a lot of brackets in the results, which make it difficult for the reader to understand the described results, please change it somehow, not always all values from the tables should be placed in the text.

Answer 6. Thank you for your suggestion. The text has been reviewed and results that were in tables were deleted, as well as others that were not essential for the understanding of the text.

  1. line 216: change to: statistically significant

Answer 7. It has been done.

  1. The discussion gives the impression of re-reading the results, please edit this part to avoid repetition of results, especially numbers, and focus on their interpretation and conclusions.

Answer 8. Thank you for your suggestion. The discussion has been edited to avoid the impression of re-reading the results. The repetitions of results have been reduced and some phrases have been moved to results.

  1. Line 298: how do authors know that MDRB strains are not more virulent than others? They did not test virulence, please remove this statement or explain on what basis they assessed virulence?

Answer 9. Thank you for your suggestion. We have deleted the term “virulence”. The new sentence reads: “Our results suggest that MDRB not necessarily have a deleterious effect, as shown in other works (68,69)” lines 326-327.

  1. Line 315 - 320:: A sentence with 6 lines and a whole paragraph is way too long and hard to understand. Please rephrase it.

Answer 10. We have rephrased it, following your indication: “For this reason, we consider that, although in this study the etiology of MDRB does not seem to determine the prognosis, in UTI in other circumstances or in other infections in which the isolation of the microorganisms is not so frequent, it may be very important to know the risk factors for MDRB. This would allow us to carry out an early adjustment of the antibiotic therapy, making it possible to reduce the effect of IEAT on mortality.” Lines 343-347”

Reviewer 2 Report

In this single center prospective observational study over a 5.5 year period, the study authors have attempted to identify the risk factors related to community-acquired sepsis as well as risk-factors associated with 30-day mortality. Their study raises some interesting points and would be a valuable addition to the literature regarding the growing threat of antimicrobial drug resistance. However, I have some comments below regarding the study definitions and their statistical analysis.

Major Comments

1. The study authors have included patients with a positive urine culture and community-acquired urosepsis in their study. What was the study definitions regarding the timeframe of positive culture testing – was it within 24 hours or within 48 hours of diagnosis? It would help having this standardized definition included in the study. The study authors have noted specific definitions related to “Community onset healthcare-associated (HCA) urinary sepsis” and “Community-acquired infection”, as well as the definitions for MDR and XDR organisms. It would be helpful to have a separate sub-heading related to “Study Definitions” for ease of readability.

2. Do the authors have information related to a prior diagnosis of UTIs in the study population, in both groups of patients? It would be helpful to include that data and see if there were any differences between the two populations. One would presume that patients with urinary sepsis due to MDRB may have had a higher incidence of prior UTIs, but comparing the data would be educative. Additionally, if there was a significant difference between the two groups, that could be evaluated as a risk factor in the multivariate analysis.

3. The study authors had noted in the Results that severe dependency (Barthel score <40), altered mental status, health-care associated urinary sepsis were significantly associated with urinary sepsis due to MDRB in the univariate analysis. Subsequently, only healthcare-associated urinary sepsis was found to be an independent risk factor by them on multi-variate analysis. However, on reviewing the characteristics of Table 1, it appears that previous hospitalization and previous antimicrobial therapy were also significantly associated with urinary sepsis due to MDRB in the univariate analysis. Was there a particular reason this was not included in the multivariate model? The authors should clarify this point.

4. The study authors note that the median age of the study participants was 82 years. Although there was no significant difference between patients with MDR infections and those with non-MDR urinary sepsis, the older age group of their study population may be another limiting factor for direct comparison to other studies and this should be discussed in the Limitations.

Minor Comments

1. Page 2 of 14, line 70 – the study authors may have inadvertently written “June” as “Juny 2022”. Please rectify as needed.

2. Page 4 of 14, lines 119-120 – please include the publishers of the statistical software [International Business Machines, Armonk, New York, USA] as per standard nomenclature.

3. The study authors have noted the risk factors for 30-day mortality in the Results (page 7 of 14, lines 199-207). For ease of readability, it would be helpful to include another Table highlighting the results of the univariate and multivariate analysis.

Author Response

Thank you to the reviewers for their comments and suggestions, which have been most helpful and useful for improving the quality of the text and the readers’ comprehension. I will answer the reviewers’ questions point by point.

 Major Comments

  1. The study authors have included patients with a positive urine culture and community-acquired urosepsis in their study. What was the study definitions regarding the timeframe of positive culture testing – was it within 24 hours or within 48 hours of diagnosis? It would help having this standardized definition included in the study. The study authors have noted specific definitions related to “Community onset healthcare-associated (HCA) urinary sepsis” and “Community-acquired infection”, as well as the definitions for MDR and XDR organisms. It would be helpful to have a separate sub-heading related to “Study Definitions” for ease of readability.

Answer 1. Thank you for your suggestion. A new definition has been included at the beginning of the sub-heading Data collection and definitions: “All patients included in this study met the diagnostic features of complicated urinary tract infection, namely: pyuria and bacteriuria on urinalysis by microscopy and at least one of the following: (i) symptoms of cystitis (dysuria, urinary urgency and/or urinary frequency) along with fever or other symptoms of systemic illness, such as chills, rigors, or acute mental status changes; (ii) flank pain and/or costovertebral tenderness; or (iii) fever or sepsis without localizing symptoms if other causes have been ruled out. Some cases were excluded when, despite meeting these criteria, the urine culture was negative in the following days.”

Furthermore, we have added three new sub-headings in the Methods section: "Study desing and patients", "Data collection and definitions" and "Statistical analysis".

  1. Do the authors have information related to a prior diagnosis of UTIs in the study population, in both groups of patients? It would be helpful to include that data and see if there were any differences between the two populations. One would presume that patients with urinary sepsis due to MDRB may have had a higher incidence of prior UTIs, but comparing the data would be educative. Additionally, if there was a significant difference between the two groups,that could be evaluated as a risk factor in the multivariate analysis.

Answer 2. Thank you for your suggestion. Unfortunately, a limitation of our study is that we did not collect the history of previous UTI or MDRB infection. It has been amended at the discussion. The new sentence reads: “Another limitation is that we did not collect the history of previous UTI or MDRB infection, which has been considered a risk factor for antibiotic resistance“ 

  1. The study authors had noted in the Results that severe dependency (Barthel score <40), altered mental status, health-care associated urinary sepsis were significantly associated with urinary sepsis due to MDRB in the univariate analysis. Subsequently, only healthcare-associated urinary sepsis was found to be an independent risk factor by them on multi-variate analysis. However, on reviewing the characteristics of Table 1, it appears that previous hospitalization and previous antimicrobial therapy were also significantly associated with urinary sepsis due to MDRB in the univariate analysis. Was there a particular reason this was not included in the multivariate model? The authors should clarify this point.

Answer 3. As you say, both previous hospitalization and previous antimicrobial therapy were significantly associated with MDRB-urinary sepsis by univariate analysis. However, these terms were not included in the multivariate analysis because both terms were criteria of health-care associated urinary sepsis. Therefore, they were not included in the logistic regression model to avoid collinearity.

  1. The study authors note that the median age of the study participants was 82 years. Although there was no significant difference between patients with MDR infections and those with non-MDR urinary sepsis, the older age group of their study population may be another limiting factor for direct comparison to other studies and this should be discussed in the Limitations.

Answer 4. This has been done. “Finally, the older age of the study participants may be a limiting factor for direct comparison to other studies carried out on younger patients.”

Minor Comments

  1. Page 2 of 14, line 70 – the study authors may have inadvertently written “June” as “Juny 2022”. Please rectify as needed.

Answer1. This has been done.

  1. Page 4 of 14, lines 119-120 – please include the publishers of the statistical software [International Business Machines, Armonk, New York, USA] as per standard nomenclature.

Answer2. This has been done.

  1. The study authors have noted the risk factors for 30-day mortality in the Results (page 7 of 14, lines 199-207). For ease of readability, it would be helpful to include another Table highlighting the results of the univariate and multivariate analysis.

Answer3. Thank you for your suggestion. We have included a new table in the manuscript: Table 5. Univariate and multivariate analysis by logistic regression of risk factors for 30-day mortality of 193 cases of community-acquired urinary sepsis  

Round 2

Reviewer 2 Report

The study authors have incorporated the suggested changes and explained their inclusion/exclusion of other points. I commend the authors for making those changes and I have no major comments to add.

I just had two minor comments as below:

1. In the Abstract, the study authors note that “the impact of MDR bacteria on the outcomes of community-acquired urinary sepsis was mild”. In the Discussion, they report that “the impact of MDRB in patients with community-acquired urinary sepsis was moderate”. It is difficult to quantify what would characterize as “mild” or “moderate” in such a situation. A better way to consider expressing this is just to summarize the main study findings. Something like the following could be considered “In conclusion, patients with MDRB in our study had similar rates of mortality and hospitalization compared to non-MDRB patients, with healthcare-associated US being noted to be an independent risk factor for MDR bacterial infection”

2. Please ensure that the same text fonts are used for Tables 4 and 5 as those used for other tables in the manuscript.

Author Response

First of all, thank the reviewer for his recommendations, and we answer each one of them. As the modifications of the first review have not yet been included in the manuscript sent by the publisher, the responses to this second review have been sent marked in blue.

In the Abstract, the study authors note that “the impact of MDR bacteria on the outcomes of community-acquired urinary sepsis was mild”. In the Discussion, they report that “the impact of MDRB in patients with community-acquired urinary sepsis was moderate”. It is difficult to quantify what would characterize as “mild” or “moderate” in such a situation. A better way to consider expressing this is just to summarize the main study findings. Something like the following could be considered “In conclusion, patients with MDRB in our study had similar rates of mortality and hospitalization compared to non-MDRB patients, with healthcare-associated US being noted to be an independent risk factor for MDR bacterial infection

Answer1.

In response to the recommendation, the conclusions have been reformulated with the following format, maintaining the structure of the abstract:

"In conclusion, patients with MDRB in our study had similar rates of mortality and hospitalization compared to non-MDRB patients, with healthcare-associated US being noted to be an independent risk factorfor MDR bacterial infection."

2. Please ensure that the same text fonts are used for Tables 4 and 5 as those used for other tables in the manuscript.

Answer 2. 

The typographic and format changes to the new table have not yet been made by the editor, so we hope they will be made in the next shipment. That is why the latest modifications of this 2 revision have been sent marked in blue.
